# Goldilocks Dilemma: LPS Works Both as the Initial Target and a Barrier for the Antimicrobial Action of Cationic AMPs on *E. coli*

**DOI:** 10.3390/biom13071155

**Published:** 2023-07-20

**Authors:** Martin Jakubec, Fredrik G. Rylandsholm, Philip Rainsford, Mitchell Silk, Maxim Bril’kov, Tone Kristoffersen, Eric Juskewitz, Johanna U. Ericson, John Sigurd M. Svendsen

**Affiliations:** 1Department of Chemistry, Faculty of Science and Technology, UiT the Arctic University of Norway, 9019 Tromsø, Norway; fredrik.g.rylandsholm@uit.no (F.G.R.); philip.rainsford@uit.no (P.R.); tone.kristoffersen@uit.no (T.K.); john-sigurd.svendsen@uit.no (J.S.M.S.); 2Department of Pharmacy, Faculty of Health Sciences, UiT the Arctic University of Norway, 9019 Tromsø, Norway; maxim.brilkov@uit.no; 3Department of Medical Biology, Faculty of Health Sciences, UiT the Arctic University of Norway, 9019 Tromsø, Norwayjohanna.e.sollid@uit.no (J.U.E.)

**Keywords:** AMP, SPR, NMR, liposomes, LPS, lipid binding

## Abstract

Antimicrobial peptides (AMPs) are generally membrane-active compounds that physically disrupt bacterial membranes. Despite extensive research, the precise mode of action of AMPs is still a topic of great debate. This work demonstrates that the initial interaction between the Gram-negative *E. coli* and AMPs is driven by lipopolysaccharides (LPS) that act as kinetic barriers for the binding of AMPs to the bacterial membrane. A combination of SPR and NMR experiments provide evidence suggesting that cationic AMPs first bind to the negatively charged LPS before reaching a binding place in the lipid bilayer. In the event that the initial LPS-binding is too strong (corresponding to a low dissociation rate), the cationic AMPs cannot effectively get from the LPS to the membrane, and their antimicrobial potency will thus be diminished. On the other hand, the AMPs must also be able to effectively interact with the membrane to exert its activity. The ability of the studied cyclic hexapeptides to bind LPS and to translocate into a lipid membrane is related to the nature of the cationic charge (arginine vs. lysine) and to the distribution of hydrophobicity along the molecule (alternating vs. clumped tryptophan).

## 1. Introduction

The rapid spread of multidrug-resistant bacteria and the slow discovery of new classes of antibiotics is leading researchers to consider alternatives to classical antibiotics. Antimicrobial peptides (AMPs) represent one promising alternative [1,2]. AMPs are a ubiquitous part of the innate immune defense in all living organisms, from humans to bacteria themselves. They have been widely studied, and today, more than 3000 natural AMPs have been reported and characterized [3,4,5]. However, most natural AMPs are ill-suited as drug candidates due in part to either poor ADME-Tox properties (absorption, distribution, metabolism, and toxicity) or overly complex chemistry [6,7]. Consequently, de novo-designed antimicrobial peptides (AMP) are being developed with simplified sequences focusing on a small number of core residues [8]. These synthetic AMPs combine intelligent design, microbiology, chemistry, and a knowledge of human metabolism in a series of iterations to provide highly efficient compounds. However, despite decades of AMP study, there is a lack of in-depth knowledge of their modes of action, including potential targets for the AMPs in live bacteria. Case in point: while a number of proposed modes have been put forth, such as pore-formation, carpeting, and detergent-like, these models are often founded on observations in simplified model systems, at concentrations that are poorly representative of biological conditions—at least with regards to antimicrobial assays [9,10].

Currently, the largest class of AMPs contains a significant fraction of cationic residues [11]. The net cationic charge of these peptides is believed to be essential to their mode of action and selectivity, with the cationic sites playing a crucial electrostatic role in the interaction between the AMPs and components of the bacterial membrane. Arginine and lysine are the most common acids in cationic AMPs, with very similar overall abundances. Still, lysine is more common in AMPs; meanwhile, arginine is more common in antiviral peptides [12,13]. Studies with synthetic cationic AMPs, either as derivatives of natural peptides or of de novo design, have unequivocally shown that arginine analogs display higher antimicrobial efficacy than corresponding lysine analogs [14,15,16]. While the effect of arginine on antimicrobial activity is well established, the underlying mechanism is not fully understood. For example, it has been observed that increased arginine content enhances peptide embedding in artificial POPC:POPG membranes [15], with the ability of arginine to stabilize its charge in the bilayer [17] or the ability of arginine to hydrogen bond with surrounding water when involved with cation-π interactions with tryptophan [17,18], being linked to arginine’s ability to embed efficiently. However, no consensus hypothesis on this phenomenon exists.

Another essential component of AMP activity is the hydrophobic volume of the peptide. The ratio of charged amino acids and hydrophobic volume is reported to correspond with the activity of AMPs on phospholipid-based model membranes, an observation suggesting that the main mode of action for cationic AMPs is bilayer disruption [19]. The antimicrobial peptide database—ADP3 shows that 80% of all entries possessing reported antimicrobial activities contain at least one of the hydrophobic phenylalanine or tryptophan [20]. The hydrophobic bulk of these residues is believed to be important to the membrane disruption by positioning in the interfacial region of the bilayer [10]. In particular, the paddle-like tryptophan leverages its unique indole moiety, forming hydrogen bonds with the lipid-tail carbonyl group, and at the same time, its aromatic bulk is able to disturb the lipid packing. Indeed, it has been previously shown that tryptophan substitution with less hydrophobic amino acids can be used to fine-tune the natural lytic ability of rationally designed peptides [19].

While the importance of the interplay between hydrophobicity and charge for membrane interaction and disruption is well ingrained in the consciousness of AMP research, what is often not considered is that the charge and lipophilic nature of AMPs are important properties for binding to the lipopolysaccharides (LPS) of Gram-negative bacteria. LPS is a major component of the outer membrane of Gram-negative bacteria and acts as a barrier that AMPs need to cross in order to reach the bacterial membrane [21,22,23]. The sequence of initial LPS binding before membrane localization or intracellular binding cannot be avoided, and thus initial LPS interaction acts as a major bottleneck for most AMPs. In this work, we have observed that the initial binding to LPS needs to be fine-tuned for cationic AMPs to attain their antimicrobial activity. Just as in Southey’s tale [24], Goldilocks needed to find her ideal conditions, so the initial interaction between AMPs and the LPS needs to be ideal. Too strong LPS binding may prevent AMPs from reaching the membrane target or cause aggregation of peptides in the cell wall [25]; LPS interaction that is too weak would reduce the amount of AMP available and prevent adequate permeation through the tight network of LPS to the bacterial membrane. In short, AMPs need to find their ‘Goldilocks’ zone of LPS affinity, where the interaction is not too tight, not too loose, but “just right”.

Here we present evidence that the interaction between AMPs and Gram-negative bacteria is driven by an initial binding to the LPS. All compounds reaching the bilayer, or an intracellular target, must first cross the LPS. In addition, we introduce a toolbox of suitable experiments that can be used to quantify this kinetic process—a combination of surface plasma resonance (SPR) and NMR experiments with live bacteria: live-cell NMR. We have tested this toolbox in the study of the interactions of a series of cyclic cationic AMPs (cAMPs) and several *E. coli* strains, including LPS-deficient and colistin-resistant strains. Using this toolbox, we can quantify the initial binding events towards several *E. coli* strains, and we are able to identify lysine as an LPS binder in the ‘Goldilocks’ zone of affinity.

## 2. Materials and Methods

All the common chemicals are of analytical purity and supplied by Merck KGaA, Darmstadt, Germany. All the lipid samples had been supplied by Avanti Lipids (Alabaster, AL, USA).

### 2.1. Peptide Synthesis

The peptides were prepared from 2-Chlorotrityl chloride resin and Fmoc-Trp(Boc)-OH as the first amino acid using a solid phase peptide synthesis strategy. The synthesis followed synthetic protocols previously reported [26]. The side-chain-protected linear peptide precursors were prepared by an automated peptide synthesizer (Biotage Initiator + Alstra). Following resin cleavage, the peptides were head-to-tail cyclized in solution, side-chain deprotected, and the crude peptides were purified by preparative reverse-phase high-performance liquid chromatography. Chemical analysis of synthesized peptides is available in the Appendix A.

### 2.2. MIC Testing

The bacterial strains tested were *E. coli* ATCC 25922, *E. coli* CCUG 70562, and *E. coli* NR-698. The MIC assays followed the CLSI guidelines [27]. Overnight cultures and MIC assays were performed in MHBII (212322, Becton, Dickson and Company, Sparks, MD, USA). In addition, *E. coli* NR-698 was tested after cultivation with 4 µg/mL of colistin to induce alternation in surface charge.

### 2.3. Liposomes Preparation

The large unilamellar liposomes of DMPC and DMPC:PG (95:5) were prepared for the SPR experiment using standard methods [28]. In brief, lipid mixtures were weighted from dry powder (Avanti, USA) and dissolved in 3:1 dichloromethane:methanol mixtures. The organic mixture was dried in a vacuum rotavapor to form a lipid film which was hydrated to 20 mM total concentration of lipids (for DMPC and DMPC:PG mixture) by adding 10 mM HEPES buffer pH 7.4 with 150 mM NaCl. For DMPC:LPS mixtures, dried LPS was dissolved in water and mixed with DMPC dissolved in dichloromethane. Isopropanol was added to the mixture until only one phase—dichloromethane:isopropanol:water was formed [29]. Dry film was prepared and then hydrated using the same method as above to a concentration of 18 mM of DMPC and 10% (*w*/*w*) LPS in HEPES buffer. Lipid liposomes were extruded 13 times through a 100 nm diameter pore using an Avanti Mini Extruder kit.

### 2.4. SPR

The SPR experiments were performed at room temperature using the L1 chip and T200 Biacore instrument (GE Healthcare, Chicago, IL, USA). The experimental setup is the same as in Jakubec et al. (2020) [30]. Briefly, lipid liposomes were immobilized on a cleaned surface of the chip using a flow rate of 2 µL/min for 2400 s. The immobilization efficiency was tested by injecting 0.2 mg/mL of bovine serum albumin (BSA, A7030, Sigma-Aldrich, Saint Louis, MO, USA) for 1 min at 30 µL/min. The change of RU less than 400 indicated sufficient coverage. Next, diluted peptides were injected over immobilized liposomes using a flow rate of 15 µL/min for 200 s with a subsequent 400 s dissociation phase. The peptide concentration was from 4 to 128 µM, except for poorly binding peptides c(LWwNKr) and c(WKWKWK), which were set from 24 to 768 µM to reach vesicle saturation. Samples were injected from low to high concentrations due to the possibility of sample retention in the liposomes. Between individual injections, liposomes were regenerated by three subsequent 30 s injections of 10 mM NaOH at a 30 µL/min flow rate. The control flow cell was treated the same way as the sample cells, except sample injections were replaced by HEPES buffer. The results were processed using in-laboratory written MATLAB scripts (MATLAB R2022a; scripts are available at [31]). K_D_ was obtained Using Equation (1) from a standard steady-state fit:(1)$$RU=\frac{{RU}_{MAX}}{1+\frac{K_{D}}{\left[P\right]}}+off,$$
where RU is measured SPR response; RU_MAX_ is RU response during saturation; [P] is peptide concentration; and off is offset of response.

K_P_ was obtained from steady-state using a method developed by Figueira et al. (2017) [32] using Equation (2):(2)$$\frac{{RU}_{S}}{{RU}_{L}}=\frac{\gamma_{L}K_{P}\frac{M_{S}}{M_{L}}{\left[S\right]}_{W}}{1+\delta \gamma_{L}K_{P}{\left[S\right]}_{W}},$$
where RU_S_ is the relative response of the solute (peptides); RU_L_ is the total lipid deposition response; γ_L_ is the molar volume of the lipids (average for mixtures); M_S_ and M_L_ are the molecular mass of the solute and lipid, respectively; and [S]_W_ is the concentration of solute in water. K_P_ is the partitioning constant, while σ is the lipid-to-solute ratio. The two latter are obtained from the fit.

The k_off_ rate was obtained from the first 200 s of dissociation using adapted formalism from Figueira et al. (2017) [32] using Equation (3). Contributions from two populations were identified, so the average k_off_ response was calculated by Equation (4).
(3)$$S_{L}\left(t\right)=\alpha e^{-k_{off},\alpha^{t}}+\beta e^{{-k}_{off},\beta^{t}}+S_{L,r},$$
(4)$$k_{off}=\frac{\alpha k_{off},\alpha +\beta k_{off},\beta }{\alpha +\beta },$$
where S_L_ is the linearised ratio of responses of the solute and lipid, which is plotted against the time of dissociation; α and β are individual populations; and S_L,r_ is the retained solute fraction.

The association rate k_on_ was obtained from experimentally measured K_D_ and k_off_ by Equation (5):(5)$$K_{D}=\frac{k_{off}}{k_{on}}.$$

### 2.5. NMR Experiments

NMR spectra were acquired on a Bruker Avance III HD spectrometer operating at 600 MHz for ^1^H, equipped with an inverse TCl cryo probe. All NMR spectra were acquired at 298 K using 5 mm tubes using standard pulse programs [31].

#### 2.5.1. Live Cell NMR

*E. coli* strains ATCC 25922, CCUG 70662, and NR 698 were grown until OD_600_ ≈ 0.6–0.8 in standard Lysogeny Broth at 37 °C from overnight colonies on agar plates. This allowed bacterial cultures to reach an exponential state of growth. CCUG 70662 was cultivated with and without 4 µg/mL of colistin in both agar and broth to allow for modification of the lipid A part of LPS [33]. After that, cells were carefully centrifuged by 6000× *g* for 10 min, the supernatant was discarded, and the cell pellet was resuspended in 10 mM Tris buffer with 100 mM NaCl and pH 7.4. The process was repeated two more times to wash out broth components and leftover colistin. Cells were then resuspended in 1 mL of Tris buffer, and OD_600_ was checked by dilution of 100 µL of the sample. The number of cells was estimated from OD_600_ of the sample using the formulation presented by Beal et al. (2022) [34]. Furthermore, lipid concentration was estimated to be 50 mmol per bacterial cell [35]. The part of samples was diluted to OD = 1 in Tris buffer, and zeta potential was measured on Malvern Zetasizer Nano (Malvern Ins., Malvern, UK). The concentrated cell sample was then titrated (from 5 to 160 µL) into the 500 µL of peptide (1 mg/mL) in Tris buffer with 5% D_2_O and ^1^H spectra was collected using double solvent suppression with excitation sculpting and presaturation (zqesgppepr) with extended delay d1 = 8 s to allow for complete relaxation of nuclei.

Spectra were processed automatically using TopSpin 4.0.8 (Bruker, Bremen, Germany) and by Matlab R2020b with Signal processing and Bioinformatics toolbox (Natick, MA, USA). The processing scripts are available at Github [31].

K_D_ was estimated from individual signal integral by using Equation (6) developed by Shortridge et al. (2008) [36] for invisible binding states:(6)$$B_{expt}=1-\frac{1}{1+\frac{{\left[Lip\right]}_{T}}{{\left[AMP\right]}_{T}+K_{D}}(\frac{\gamma_{B}}{\gamma_{F}}-1)},$$
where B_expt_ is measured integral decrease; [Lip]_T_ is total lipid concentration estimated from the amount of titrated cells; [AMP]_T_ is total peptide concentration; and γ_B_ and γ_F_ are the line widths of bound and free AMP, respectively. Due to the nature of interaction and the invisibility of the bound state line, the width ratio was estimated from the fit, and it was used for all proton signals. This value was different for each combination of AMP and bacterial sample, and it is noted in the Appendix A.

#### 2.5.2. Assignment

NOESY, TOCSY, DQF-COSY, ^15^N-HSQC, HSQC, HMBC, and H2BC NMR spectra were recorded with standard settings. NOESY and TOCSY spectra were recorded with mixing times of 100–400 ms and 80 ms, respectively. Spectra were processed with TopSpin 3.6.0, and assignments were done manually. The full assignment is reported in the Appendix A.

## 3. Results and Discussion

### 3.1. Cyclic Peptides with Balanced Lipophilic Bulk and Cationic Charge

In the present work, we have selected to limit the conformational freedom the peptides may adopt with head-to-tail cyclization. The peptides designed for this study were cyclohexapeptides with an equal number of cationic residues (Arg or Lys) and bulky lipophilic residues (Trp), thus fulfilling the pharmacophore for short synthetic cationic antimicrobial peptides against both Gram-positive and Gram-negative bacteria [26]. Two types of sequences were selected, either sequence where the cationic residues and tryptophan are alternating or a “clumped” sequence where the cationic residues are adjacent in the sequence and the tryptophan residues likewise. In addition, we have included a fifth peptide with a similar general structure—a cyclohexapeptide which proved to be both a very poor antimicrobial and lipid binder (Figure 1).

Cyclic hexapeptides tend to adopt conformations where two opposing β-turns link two short antiparallel β-strands. Using the Dunitz–Waser concept [27], there are two gross variants of the cyclic hexapeptide; one resembling the chair conformation of cyclohexane through the twisted βI turns, the other resembling the boat conformation of cyclohexane due to the presence of the flatter βII turns (Figure 1). An alternating sequence c(WR)3 or c(WK)3 would, due to the internal symmetry, give rise to only one structure within each gross conformation, whereas the clumped sequences, c(WWWRRR) and c(WWWKKK), would possibly form two different structures within each gross conformational category.

Moreover, the linear variants of selected peptides have already been shown to possess interesting antimicrobial potential. The linear peptide WWWRRR has been demonstrated to exhibit the best combination of length and activity [14]. Additionally, the peptide WRWRWR had demonstrated promising activity against methicillin-resistant *S. aureus* [37], while the peptide WKWKWK exhibited decreased activity when the sequence was shifted to KWKWKW [38]. Presented peptides thus correspond to new cyclic variants of linear antimicrobial peptide analogs.

**Figure 1 biomolecules-13-01155-f001:**
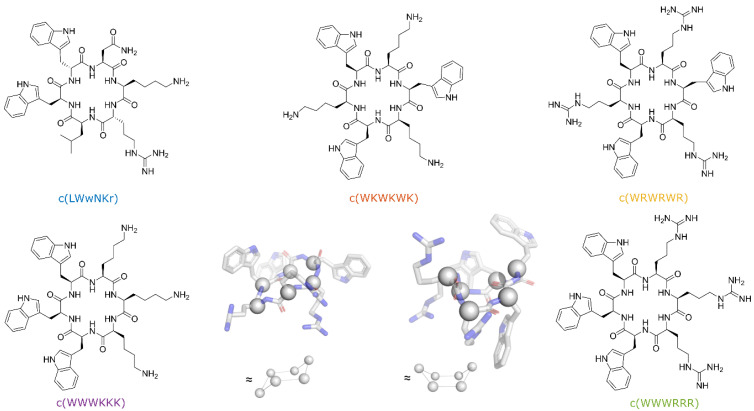
Structures of the synthesized peptides. Snapshots of c(WWWRRR) generated from CREST (provided by Prof. Kenneth Ruud) calculations showcasing the Dunitz–Waser concept [39]. Chair conformation (**left**) and boat conformation (**right**) with the alpha carbons outlined as spheres for a better overview.

### 3.2. Antimicrobial Activity of Peptides

The antimicrobial activity of the five cyclic cationic AMPs was determined against *S. aureus, P. aeruginosa*, and three strains of *E. coli*: ATCC 25922, LPS colistin-resistant strain CCUG 70662, and LPS deficient strain NR 698 (Table 1). CCUG 70662 was cultivated in two conditions, both in the absence (−) and in the presence (+) of colistin (4 µg/mL). Resistance development can be monitored by measuring changes in the ζ-potential, reflecting the overall charge of the bacteria. Indeed, cultivating CCUG 70662 in the presence of colistin caused a decrease in the ζ-potential [33]. This change is due to the addition of phosphoethanolamine to the lipid A moiety of LPS, reducing the overall negative charge of the LPS [33,40]. In the NR 698 strain, translocation of LPS from the periplasmic space to the outer leaflet is hindered. This increases the susceptibility of the bacteria to bile salts, chlorobiphenyl-vancomycin, vancomycin, bacitracin, novobiocin, rifampicin, and erythromycin, amongst others, suggesting that the poor LPS translocation results in a deficient outer membrane [41]. The most active AMPs were those with clumped residues, c(WWWRRR) and c(WWWKKK), followed by the peptides with alternating sequences, c(WRWRWR) and c(WKWKWK). Subsequently, the AMP c(LWwNKr) was included as a negative control as it showed no detectable antimicrobial activity on *E. coli* strains except the NR 698 strain and had only a weak affinity towards lipid membranes during the preliminary experiments.

One important observation is that the colistin-resistant strain CCUG 70662 is still susceptible to the tested hexapeptides even after cultivation with colistin which induced the modification of lipid A. Colistin has been shown to bind LPS and displace divalent cations from lipid A. This will increase the fluidity of LPS and cause major disruption of the outer membrane, and allows for the colistin to reach the periplasmic space [42,43]. The following bacterial toxicity is not fully explained, as outer membrane integrity is not crucial for bacteria survival. Currently, there are three major theories on colistin antimicrobial activity: direct disruption of the inner membrane, vesicle-vesicle contact pathways, or targeting of lipid A in the periplasmic space [42,43,44]. Regardless, these theories lead to the final disruption of inner membrane integrity and subsequent cell death. No change in activity against CCUG 70662+ with modified lipid A suggests that the mode of action of the tested hexapeptides is not directly related to lipid A binding. Therefore, following these MIC results, we have utilized surface plasmon resonance experiments to directly measure the compounds’ ability to translocate into lipid bilayers.

**Table 1 biomolecules-13-01155-t001:** MIC and ζ potentials for the tested peptides against *P. aeruginosa*, *S. aureus*, and the three *E. coli* strains: ATCC 25922, LPS deficient—NR 698, and colistin-resistant CCUG 70662, which was tested after cultivation with (+) and without (−) 4 µg/mL of colistin, which caused a decrease of negative charge of the cell wall. MIC values for *P. aeruginosa*, *S. aureus,* and *E. coli* ATCC 25922 are reproduced from [45].

AMP	MIC (µg/mL)					
	*P. aeruginosa* ATCC 27853	*S. aureus* ATCC 9144	*E. coli* ATCC 25922	*E. coli* CCUG 70662−	*E. coli* CCUG 70662+	*E. coli* NR 698
Zeta potential	-	-	−21.3 ± 0.6 mV	−21.8 ± 0.8 mV	−14.3 ± 0.5 mV	−21.2 ± 0.6 mV
c(LWwNKr)	>250	>250	>250	>250	>250	32
c(WKWKWK)	>250	128	64	64	64	4
c(WRWRWR)	64	32	32	16	32	2
c(WWWKKK)	32	32	8	4	4	2
c(WWWRRR)	16	4	8	8	8	4

### 3.3. SPR and Choice of Lipid Models

SPR is emerging as the method of choice for measuring the partitioning of small compounds into lipid bilayers [32,45,46]. In SPR, liposomes are immobilized on the surface of the SPR chip, and AMPs are injected over the lipid layer at increasing concentrations. The measured change in the response of the plasmon wave allows for the determination of both the partitioning, K_P_, which is defined as the concentration of peptide in the lipid bilayer over the concentration of peptide in water ([AMPs] in lipid/[AMPs] in water), and the dissociation rate k_off_ [32]. In addition, the dissociation constant (K_D—_Appendix A) can be estimated from the steady-state data using a generalized binding model. From the estimated K_D_ and the measured k_off_ values, the association rate k_on_ can be calculated (Equation (5)). For the lipid models, DMPC and DMPC:DMPG (95:5) liposomes, as well as DMPC liposomes containing 10% LPS (isolates from *E. coli* O111:B4), were used. The rationale behind the selection of models is their representation of key membranes. The DMPC liposomes possess a neutral membrane, akin to a eucaryotic membrane, and a useful probe of the lipophilic nature of the AMPs in the membrane interaction. The presence of 5% DMPG in the DMPC:DMPG liposomes provide a net anionic lipid surface, more similar to a procaryotic membrane. Finally, the LPS-containing liposomes are used to model the outer membrane of Gram-negative bacteria, with LPS being a unique and abundant component of the Gram-negative outer membrane.

### 3.4. Antimicrobial Activity Broadly Correlates with K_P_ for DMPC Liposomes

The measured K_P_ for the different peptide and lipid membrane combinations spans a wide range of values covering three orders of magnitude (Figure 2). c(WWWRRR) strongly partitions into the DMPC lipid bilayer, with the highest K_P_ of the series—6649, whereas the negative control peptide, c(LWwNKr), showed a very poor preference for DMPC with a K_P_ of 278. Introducing 5% of a negatively charged component (DMPG) to the liposomes doubled the K_P_ of all peptides except c(WKWKWK), where the increase was less substantial.

The antimicrobial activity against Gram-positive S. *aureus* closely follows the K_P_ in DMPC liposomes. However, the trend for MIC against Gram-negative bacteria was less clear. The K_P_ is calculated from steady-state conditions, so the value corresponds to the sum of possible interaction states, including “coating” of the surface membrane, self-aggregation, and any penetration. This sum of values does not correspond closely with antibacterial activity against Gram-negative bacteria. This could be due to the hindrance of peptide translocation through LPS-loaded outer membranes of Gram-negatives [47]. Despite this, K_P_ can still be used for activity ranging against Gram-negatives, as the most active peptides have K_P_ values up to one order of magnitude higher than the inactive peptides.

### 3.5. The Initial Peptide:Membrane Association Is Affected by LPS

Both c(WWWRRR) and c(WWWKKK) display significantly different K_P_ and k_off_ values when measured against DMPC and DMPC:PG liposomes, a finding that was unexpected due to their similar MIC against *E. coli*. As such, further explorations of the model space were warranted to achieve a better model for the Gram-negative. To this end, 10% LPS was incorporated into the DMPC liposomes. LPS is a crucial component of the outer membrane of Gram-negative bacteria and is absent in Gram-positive bacteria. The ketodeoxyoctanic acid component in LPS confers an anionic charge to the DMPC:LPS construct [48]. We have previously observed that the binding efficacy of the peptides was significantly enhanced by the introduction of negatively charged head groups, pointing to the importance of negative charge in the membrane for binding the cationic peptides. When 10% LPS was introduced to the DMPC membranes, enhanced binding of the AMPs to the DMPC:LPS construct was observed—similar to the introduction of DMPG to the DMPC membranes.

All active peptides demonstrated an increase in K_P_ to the DMPC:LPS membranes compared to DMPC. A greater increase in binding to the LPS membrane relative to the negatively charged DMPC:PG membranes was also generally observed. However, two noteworthy exceptions can be identified. Firstly, c(LWwNKr) sees a four-fold increase in K_P_ compared to the DMPC and DMPC:PG constructs, suggesting a strong preference for the LPS. Secondly, although c(WWWKKK) sees an enhanced K_P_ when LPS is present, this effect is less pronounced than in DMPG:DMPC membranes. This indicates that the anionic charge of the lipid head group is a greater driver of the interaction between the AMP and the lipid bilayer than LPS. The presence of LPS has no effect on k_off_ for any of the active peptides. Instead, LPS significantly increases their association rate k_on_. This is coherent with the proposal that LPS is a driving force for the initial interactions of AMPs but that it is not the final destination; this remains the lipid bilayer. Hence, the introduction of LPS in the bilayer boosts k_on_ while k_off_ remains largely unchanged.

LPS forms a unique combination of two barriers: a hydrophilic barrier is provided by a densely packed oligosaccharide core, and a hydrophobic barrier is provided by the hydrocarbon chain region of lipid A buried in the phospholipid membrane [49]. During the initial interaction between AMPs and LPS, the oligosaccharides can hinder the cationic AMPs and slow their localization into the membrane. This has been shown previously with the linear peptide K_5_L_7_, where its association with LPS prevents it from reaching its target, reducing its activity [21]. Meanwhile, its diastereomer 4D-K_5_L_7_, which can penetrate LPS, has higher activity. This modification of peptide configuration changes its binding mode with LPS without affecting the lipophilicity of the peptide and renders one peptide active and the other inactive. The difference in K_P_ for the clumped c(WWWRRR) and c(WWWKKK) is not reflected in their antimicrobial activity and can be interpreted as a consequence of the same LPS-induced phenomenon. In the presence of LPS, a much stronger association of c(WWWRRR) than c(WWWKKK) to the DMPC:LPS liposomes can be observed. If the activity was to follow affinity directly, c(WWWRRR) would be expected to be the most active of the two; however, both peptides have similar MIC values across all strains. This suggests that the antimicrobial activity of c(WWWRRR) is hindered by LPS association; that is to say, it is bound too tightly and strays out of the Goldilocks zone of LPS binding. The loss of activity in the presence of LPS can also be observed on the less lipophilic peptides (with lower K_P_) c(LWwNKr), c(WKWKWK), and c(WRWRWR) as the MIC of these peptides is improved against NR-698. This strain has deficient LPS transport to the outer membrane, and as a result, there is a lower abundance of LPS in the outer membrane, which is replaced by phospholipids [41].

The SPR results, taken as a whole, suggest that peptides with a good combination of hydrophobicity and hydrophilicity, translating into a beneficial combination of k_on_ and k_off_ rates, will pass the LPS and outer membrane unperturbed reaching their antimicrobial target—either the inner membrane or an intracellular binding partner. From the tested peptides, the best combination of these rates is observed for c(WWWKKK)—a high k_on_ towards LPS allows efficient entry, but a relatively high k_off_ allows for translocation towards the inner bilayer and prevents peptide sequestration in the outer membrane. Indeed, of the tested AMPs, c(WWWKKK) seems to have k_off_ and k_on_ rates within the most efficient combined range, following the Goldilocks principle. However, lysine is known to be generally less destructive towards lipid bilayers [17].

### 3.6. NMR Line Shape Analysis Can Be Used to Probe Binding Strength of AMPs towards Lipid Liposomes

In addition to SPR, 1D ^1^H line shape NMR analysis was performed by titration of lipid liposomes into a fixed concentration of peptide. The size of the vesicle/AMP complex will result in line broadening and signal loss of the monitored compound with an increasing concentration of liposomes. This loss of the signal can be quantified, and the dissociation constant, K_D_, can be estimated [36,50]. This method has previously been used to monitor the kinetics of enzymes and various inhibitors [51], to monitor protein binding to different lipid models [30], or in the qualitative assessment of AMP binding towards bacterial cells [52]. However, it is important to note that signal attenuation and line broadening are caused by relaxation, sample inhomogeneity, as well as contributions from the chemical exchange in the NMR time scale, which can vary for different molecules, different interactions, and different concentration ranges. Nevertheless, SPR showed that all five tested peptides have relatively slow off-rates, k_off_ < 10 s^−1^ [53]. If we assume that the off-rates for free liposomes in solution are comparable to those of immobilized and partially fused liposomes to the SPR chips [32], then the contributions from on/off-exchange processes are not expected to significantly affect the effective relaxation, and the major source of attenuation is fast relaxation caused by the slow correlation time of the peptides in complex with the lipid bilayer or LPS (Figure 3).

The results from the line shape analysis can be used to rate the peptides in the same order as SPR from highest to lowest affinity: c(WWWRRR) > c(WWWKKK) ≫ c(WRWRWR) > c(WKWKWK) > c(LWwNKr). When LPS was present, the c(LWwNKr) and c(WKWKWK) peptides were most affected by the change, with a notable decrease in K_D_. The most efficiently bound peptides, c(WWWRRR) > c(WWWKKK), were not affected by the LPS addition. The NMR line shape analysis data follows the same general qualitative trend of the SPR with minor quantitative differences, which could be caused by the different states of the lipid bilayer between the NMR and SPR experiments. Nevertheless, line shape analysis offers a simple, alternative method to explore peptide binding towards model lipid membranes.

### 3.7. Live Cell NMR Highlights Strain Differences

While binding of the AMPs is a necessity to achieve bacterial killing, it may neither be sufficient to explain the differences between the activities of individual compounds nor the main factor for the mode of action. We aimed to expand the NMR toolbox to include live cells with the aim of connecting the observed lipid partitioning, MIC activities, and mode of action. To our knowledge, this is the first time the live cell NMR approach has been used to quantify peptide:bacterial cell binding.

For the live cell NMR, the same *E. coli* strains as for the MIC testing (see above) were used. In addition, the ATCC 25922 cell lysate was used to observe the effect the release of intracellular components has on peptide binding. To measure the initial AMP:bacterial cell binding, we titrated an increasing number of cells into a known and fixed concentration of AMP and monitored the signal attenuation of the peptide (Figure 4A,B). From this, we have extracted the K_D_ for each AMP towards the bacteria using the method developed by Shortridge et al. (2008) [36] (Figure 4C), which allows us to estimate the binding affinity based on signal broadening (see Section 2). There were major affinity differences for individual strains tested, as summarized in Figure 4D.

### 3.8. Inactive Peptide c(LWwNKr) Binds Strongly to ATCC 25922 and NR 698

No direct correlation between the live cell affinity and MIC activity is seen in the data. Furthermore, there were significant differences between the different *E. coli* cell lines, highlighting AMP:strain specificity. Accordingly, a conservative interpretation of the differences in binding affinity between different strains is called for. However, interesting observations of the differences in AMP binding within each individual strain exist. For example, the SPR results showed that peptide c(LWwNKr) has poor partitioning and retention in a lipid bilayer, which is only slightly improved by the presence of LPS. Now, the markedly different affinity of c(LWwNKr) to different strains suggests that this peptide can interact with an unknown component (possibly membrane proteins) in the outer part of the bacterium, different than phospholipids and LPS, which is more abundant in ATCC 25922 and NR 698 and less present or mutated in CCUG 70662. In addition, the K_D_ is decreased for the lysate ATCC 25922 sample, suggesting non-specific binding with intracellular components (larger surface area) or increased availability of alternate binding sites. Still, binding of the c(LWwNKr) peptide provides no or very inefficient antimicrobial effect, as shown by its high MIC values.

### 3.9. Active Peptides Are Less Affected by Changes in Membrane Composition

There is no significant change in K_D_ for c(WWWRRR), c(WWWKKK), and c(WKWKWK) between the lysate and live cell ATCC 25922. This suggests that these peptides are able to interact with their target in intact cells and are not affected by the presence of free intracellular components. Meanwhile, with c(WRWRWR), there is a two-fold increase in K_D_ (a decrease in affinity after sonication). This may be caused by the partial disintegration of LPS molecules by sonication [54], and so eliminating the major binding site of c(WRWRWR).

The preference of c(WRWRWR) towards LPS can also be seen from a decreased affinity towards the activated CCUG 70662+, where lipid A is modified, compared to the native CCUG 70662−, highlighting the impact of the charged component of LPS on the binding of c(WRWRWR). The decrease in affinity may also be accompanied by a decrease in antimicrobial activity, with a single titer increase in MIC against CCUG 70662+ (Table 1). Taken together, this may indicate that c(WRWRWR) derives its activity by binding to the lipid A component of LPS. An increase in K_D_ is also observed in c(WKWKWK) when lipid A is modified, though this is less pronounced in comparison to c(WRWRWR). In opposition to this trend, both clumped peptides had decreased K_D_ when lipid A was modified. These results point to the main driving force of clumped peptides being hydrophobic interactions, while the alternating peptides are driven by charge. The SPR results support the observation that arginine is a more proficient LPS binder than lysine, suggesting that arginine-containing peptides should be most affected by the LPS modification. c(WWWRRR) is able to compensate for the reduced LPS affinity with increased hydrophobicity, which allows the peptide to efficiently partition into the lipid bilayer. However, this advantage is lost in the alternating peptide c(WRWRWR), and this is evidenced by the large increase in K_D_ towards the modified LPS strain.

Due to strain differences, it is not possible to compare ATCC 25922 and NR 698 directly. However, c(WRWRWR) demonstrated K_D_ differences between the two strains that were not evident for the other peptides. For ATCC 25922, the K_D_ of c(WRWRWR) is in a similar range to the other peptides (~50–200 µM), and for NR 698, it moves beyond 500 µM. This could be explained by the decrease of LPS in the outer membrane of NR 698. The notion that c(WRWRWR) binds lipid A may be further strengthened by the fact that there is reduced affinity towards the NR-698 strain, as the reduced transport of LPS to the outer membrane results in lower availability of lipid A [41].

### 3.10. Different Binding Modes of Hexapeptides towards ATCC 25922

The protons of most of the tested peptides experienced similar attenuation through the whole peptide; this led to uniform K_D_ values, as shown in Figure 5D. This is especially true for peptides with alternating sequences where due to their molecule symmetry, we can only observe the average proton signal for dipeptides units (WR or WK). However, some combinations of the clumped peptides and strains showed outliers for specific protons. These outliers suggest that there is a shift in the average binding of the peptide, i.e., a preference for a particular orientation/conformation of the peptide or a change in the internal dynamics of the ring. A breakdown of these outliers for binding towards ATCC 25922 is in Figure 5. For c(WWWKKK), the highest K_D_ was observed for the protons on Cβ of the three tryptophan residues (Figure 5E). Meanwhile, the highest K_D_ for c(WWWRRR) was localized on Cβ of Arg2 and Cγ of Arg3 (Figure 5F). This shows that, on average, there is a markedly different type of interaction between these two peptides and the bacteria. This deviation could be due to either structural differences or different interaction dynamics of the peptides—suggesting differences in binding modes.

Arginine and lysine are both cationic residues with very similar properties. They are both basic amino acids with similar pK_A_ in an aqueous environment (∼12.5 for Arg [55] and ∼10.5 for Lys [56]), which enables them to be charged in most physiological circumstances. It has been shown that these amino acids can be, to some extent, exchangeable without affecting the peptide’s secondary structure [15,57]. However, it has also been shown that Arg is more lipophilic as poly-Arg can cross cells more efficiently than oligomers of Lys [58]. In fact, the substitution of Arg for Lys has been shown to decrease the translocation efficiency of host defensive peptides [59,60]. Li et al. (2013) [17] showed that this enhanced lipophilicity could be explained by Arg’s ability to maintain its diffused charge even in the bilayer, unlike Lys, which seems to cross the bilayer deprotonated, or by the ability of Arg to form hydrogen bonds and cation-π interaction at the same time [18].

This difference in the nature of the arginine and lysine interactions with the lipid bilayer could explain some differences in signal attenuation observed in the live cell NMR. Where the poorer affinity of Lys will cause c(WWWKKK) to bind with preferable sidechains, meanwhile c(WWWRRR) will quickly orient itself in the bilayer, burying its hydrophobic groups with the Arg chains available for exchange closer to the lipid headgroups; this will lead to the higher disruption effect of arginine. Arg has a strong killing efficacy; however, it can be sequestrated in the LPS layer [61]. Opposingly, Lys is able to reach the bilayer unhindered, though it is not as efficient in its disruption. The combination of these effects could lead to the similar antimicrobial activity of c(WWWKKK) and c(WWWRRR).

## 4. Conclusions

Using lipid model systems, we have observed a correlation between AMPs amino acid composition and its binding towards DMPC liposomes and its correlation of lipid affinity with antimicrobial activity. Arginine has shown to be a more efficient lipid binder than lysine (c(WWWRRR) > c(WWWKKK)), and the clumped hydrophobic bulk of peptides formed by tryptophan showed to be more efficient than alternating sequences (c(WWWRRR) > c(WRWRWR)). This trend was further enhanced when we introduced 5% of DMPG in liposomes, highlighting the increased selectivity of cationic peptides towards anionic membranes.

However, when moving from these simple models to more complex bacterial membranes, the complexity of interactions radically increases, and the simple correlation of binding = activity is lost. The presence of only one additional component, LPS, shows drastic changes in AMPs rating, partially explaining why peptides c(WWWRRR) and c(WWWKKK) have different lipid partitioning but the same MIC for Gram-negative bacteria; LPS efficiently binds to c(WWWRRR), lowering its potential.

Nevertheless, it is within live cells where the activity takes place, and these environments should be explored further. Live cell NMR (line shape analysis on live bacterial cells) could help bridge this gap and offer a simple tool to measure complex interaction processes. It has already helped to identify several potential binding partners, such as c(WRWRWR) and lipid A binding or c(LWwNKr) and protein interactions. Further investigations are required to fully understand the mode of action of these AMPs, but the identified targets provide a starting point.

## Figures and Tables

**Figure 2 biomolecules-13-01155-f002:**
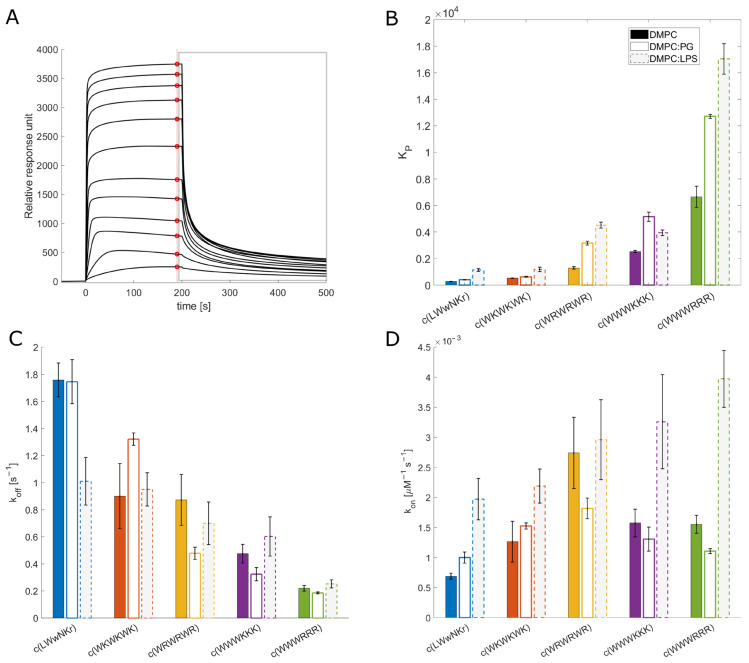
SPR results (**A**): Example SPR traces from c(WWWRRR) interaction (from 4 to 128 µM) with DMPC:PG liposomes. Red points indicate values used for steady-state fitting (to obtain K_D_ and K_P_). The grey rectangle indicates the dissociation process which is used to obtain k_off,_ according to Figuera et al. (2017) [32]. (**B**): Partitioning constants K_P_. Full, empty, and grey bars correspond to DMPC, DMPC:PG (95:5), and DMPC:LPS (90:10) liposomes, respectively. (**C**): Dissociation rate k_off_. (**D**): Association rate k_on,_ which is obtained by calculation using measured K_D_ (Appendix A) and k_on_. Values used in the plot are available in Appendix A. Values for DMPC and DMPC:PG interaction are reproduced from [45].

**Figure 3 biomolecules-13-01155-f003:**
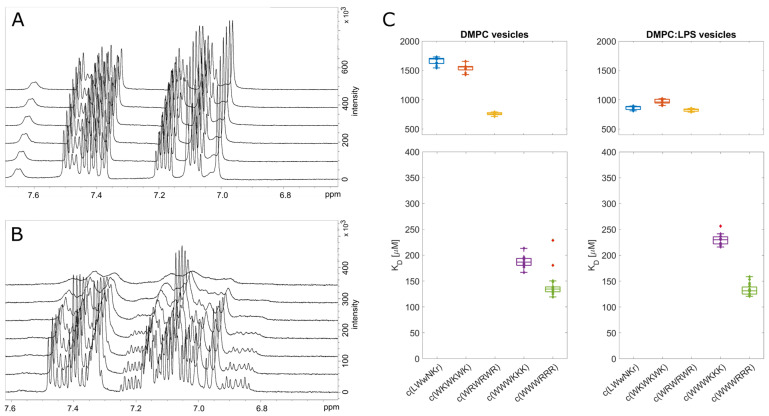
Line shape analysis of liposomes and AMP interaction. On the left are ^1^H spectra of c(LWwNKr) (**A**) and c(WWWRRR) (**B**) after titration of DMPC liposomes (from 0 to 655 µM). The signal loss (attenuation) can then be used to quantify K_D_ (**C**). Stars in the box plots indicate outliers.

**Figure 4 biomolecules-13-01155-f004:**
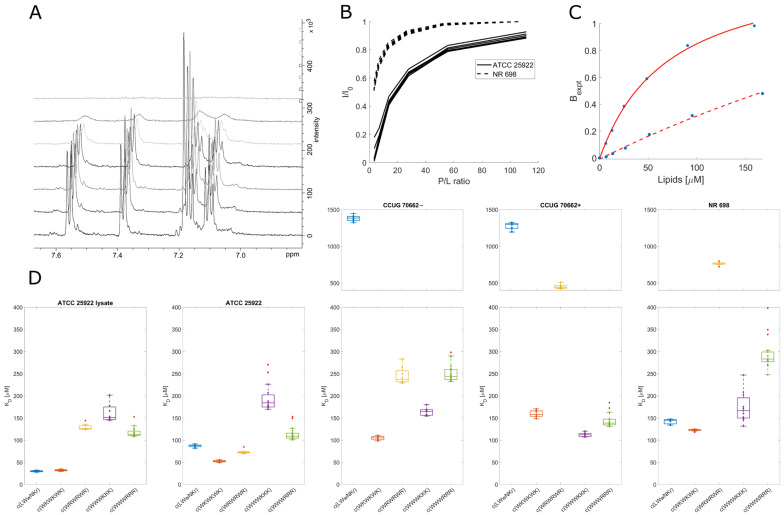
Peak broadening and signal loss during titration of *E. coli* live cells into AMPs. (**A**): ^1^H NMR spectra of c(WRWRWR) during titration of *E. coli* ATCC 25922 (increasing cell concentration from bottom to top). (**B**): Changes in proton integrals for c(WRWRWR) during titration of *E. coli* ATCC 25922 and NR 698. During titration peptide:lipid ratio decreases with the addition of more cells. (**C**): Example of fit (according to Shortridge et al. (2008) [36]) of attenuation for tryptophan ^1^H signal ζ3 of c(WRWRWR) in the presence of ATCC 25,992 (solid) or NR 698 (dashed). The assignment is included in Appendix A. The concentration of lipids per bacterial cell is estimated to be 50 mmol [35]. (**D**): Overview of K_D_ obtained from live cell NMR. For the sonication method of ATCC 25922, see Appendix A. CCUG 70662, a colistin-resistant strain, was grown without (−) and with (+) 4 µg/mL of colistin, which caused modification of lipid A and change of overall zeta potential (Table 1). NR 698 is strain deficient in LPS transport to the outer membrane, which causes membrane deficiency. Stars in the box plots indicate outliers.

**Figure 5 biomolecules-13-01155-f005:**
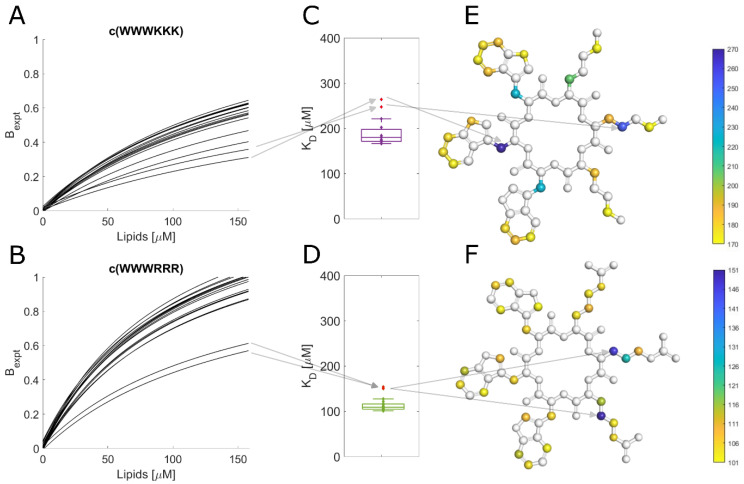
Comparison of c(WWWKKK) and c(WWWRRR) binding to ATCC 25922. (**A**,**B**): Fit of signal attenuation for all resolved proton traces according to Shortridge. (**C**,**D**): Box plot of K_D_s obtained from the fit. Arrows highlight outliers from previous proton integrals. (**E**,**F**): Structures of peptides color-coded according to measured K_D_. Arrows highlight the position of outliers: Cβ of Trp2 and Cγ of Lys2 for c(WWWKKK) and Cβ of Arg2 and Cγ of Arg3 for c(WWWRRR).

## Data Availability

All processed data can be found in the paper and Appendix A. The scripts utilized for data processing are available at [31]. Raw data can be obtained upon request.

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
