# Peer review of "Goldilocks Dilemma: LPS Works Both as the Initial Target and a Barrier for the Antimicrobial Action of Cationic AMPs on E. coli"

_biomolecules, 2023, doi:10.3390/biom13071155_

Round 1
Reviewer 1 Report
In this work, the authors have applied SPR and interestingly live-cel NMR to characterize the action of model cationic AMPs on E. coli wiht a specific focus on the role that LPS plays. This is a topic of long-standing interest and indeed discussion within the community especially in the context of how to best characterize the effects and the challenges of moving between model membrane systems and more rigourous native cellular models. The work is nicely conducted and the SPR data are compelling. I was certainly intrigued to see the live-cell NMR work because this certainly gets one much closer to understanding what is happening. I only have a few questions
(1) the SPR data are useful but it would be nice to see some images of what the liposomes look like on the SPR chip (or analogous surface) to get a sense of the structures that form. there is always that nagging question about whether they have fused, formed islands, have domains, etc...
(2) line 269 - should "compounds" be "compound's"? or "compounds' " - because it is referring to the ability so possessive context but wasn't sure if it was meant to refer to a singular compound or the compounds (plural)
(3) can the authors comment on whether there were any changes in local fluidity upon AMP binding? (esp domain-specific?)
(4) line 521 - "observe" -> "observed"
Reviewer 2 Report
Jakubek et al report a comprehensive study on the interaction between cyclic cationic amphiphilic peptides of different constitution with model lipid membranes containing lipophilic polysaccharides in an attempt to delineate the factors which determine peptide antimicrobial activity and strength of interaction. The authors used SPR and NMR techniques judicially. The work is well-planned and gives several new insights into the behavior of such systems.
I have only a couple of comments.
1. It would be nice to present similar studies on analogous linear peptides for comparison.
2. Interactions constants between the amphiphilic peptides and liposomes may be compared to those from ITC experiments in solution.
